# PEDOT-Carbon Nanotube Counter Electrodes and Bipyridine Cobalt (II/III) Mediators as Universally Compatible Components in Bio-Sensitized Solar Cells Using Photosystem I and Bacteriorhodopsin

**DOI:** 10.3390/ijms23073865

**Published:** 2022-03-31

**Authors:** Alexandra H. Teodor, Stephanie Monge, Dariana Aguilar, Alexandra Tames, Roger Nunez, Elaine Gonzalez, Juan J. Montero Rodríguez, Jesse J. Bergkamp, Ricardo Starbird, Venkatesan Renugopalakrishnan, Barry D. Bruce, Claudia Villarreal

**Affiliations:** 1Graduate School of Genome Science and Technology, University of Tennessee at Knoxville, Knoxville, TN 37996, USA; ateodor@vols.utk.edu; 2Escuela de Ciencia e Ingeniería de Materiales, Instituto Tecnológico de Costa Rica, Cartago 30101, Costa Rica; stephmonmart@estudiantec.cr (S.M.); dariana.ar@gmail.com (D.A.); tamesalexandra@estudiantec.cr (A.T.); 3Centro de Investigación y Extensión en Ingeniería de Materiales (CIEMTEC), Instituto Tecnológico de Costa Rica, Cartago 30101, Costa Rica; 4Maestría Ingeniería en Dispositivos Médicos, Instituto Tecnológico de Costa Rica, Cartago 30101, Costa Rica; 5Department of Chemistry and Biochemistry, California State University Bakersfield, Bakersfield, CA 93311, USA; rnunez16@csub.edu (R.N.); egonzalez68@csub.edu (E.G.); jbergkamp@csub.edu (J.J.B.); 6Escuela de Ingeniería Electrónica, Instituto Tecnológico de Costa Rica, Cartago 30101, Costa Rica; jjmontero@itcr.ac.cr; 7Centro de Investigación y de Servicios Químicos y Microbiológicos (CEQIATEC), Instituto Tecnológico de Costa Rica, Cartago 30101, Costa Rica; rstarbird@itcr.ac.cr; 8Escuela de Química, Instituto Tecnológico de Costa Rica, Cartago 30101, Costa Rica; 9Children’s Hospital, Harvard Medical School, 4 Blackfan Circle, Boston, MA 02115, USA; v.renugopalakrishnan@northeastern.edu; 10Department of Chemistry and Chemical Biology, Center for Renewable Energy Technology, Northeastern University, 317 Egan Center, Boston, MA 02138, USA; 11Department of Biochemistry, Cellular, and Molecular Biology, University of Tennessee at Knoxville, Knoxville, TN 37996, USA; 12Chemical and Biomolecular Engineering Department, University of Tennessee at Knoxville, Knoxville, TN 37996, USA

**Keywords:** bio-sensitized solar cells, photovoltaics, PEDOT, carbon nanotubes, photosystem I, bacteriorhodopsin, biocompatible

## Abstract

In nature, solar energy is captured by different types of light harvesting protein–pigment complexes. Two of these photoactivatable proteins are bacteriorhodopsin (bR), which utilizes a retinal moiety to function as a proton pump, and photosystem I (PSI), which uses a chlorophyll antenna to catalyze unidirectional electron transfer. Both PSI and bR are well characterized biochemically and have been integrated into solar photovoltaic (PV) devices built from sustainable materials. Both PSI and bR are some of the best performing photosensitizers in the bio-sensitized PV field, yet relatively little attention has been devoted to the development of more sustainable, biocompatible alternative counter electrodes and electrolytes for bio-sensitized solar cells. Careful selection of the electrolyte and counter electrode components is critical to designing bio-sensitized solar cells with more sustainable materials and improved device performance. This work explores the use of poly (3,4-ethylenedioxythiophene) (PEDOT) modified with multi-walled carbon nanotubes (PEDOT/CNT) as counter electrodes and aqueous-soluble bipyridine cobalt^II/III^ complexes as direct redox mediators for both PSI and bR devices. We report a unique counter electrode and redox mediator system that can perform remarkably well for both bio-photosensitizers that have independently evolved over millions of years. The compatibility of disparate proteins with common mediators and counter electrodes may further the improvement of bio-sensitized PV design in a way that is more universally biocompatible for device outputs and longevity.

## 1. Introduction

Current annual energy demands worldwide are increasing greatly, and current technologies based on carbon fuel combustion has led to significant increases in atmospheric CO_2_ levels worldwide [1]. Renewable energy harvesting from the environment, such as solar energy conversion through photovoltaic devices (PVs), is an attractive alternative that aims to harness a portion of the nearly 175,000 terajoules of solar photonic energy striking the Earth annually [2,3]. Current PV technologies on the market are made of inorganic materials including toxic heavy metals and rare-earth elements, and have prohibitive manufacturing processes, costs, and limited mobility post-installation. Furthermore, current first-generation PVs are estimated to generate nearly 80 million metric tons of waste by 2050 due to the specific recycling requirements [4]. As such, it is clear that further improvement on current PV technologies is needed to address future energy demands. Bio-sensitized solar cells (BSSCs) offer a promising way of addressing these issues using simple manufacturing processes with non-toxic and earth-abundant elements. BSSCs are based on dye-sensitized solar cells first developed by Grätzel and O’Regan, which pioneered the design of a photosensitizing dye incorporated into a thin-layer style device utilizing much more relaxed manufacturing requirements and the ability to incorporate many different types of materials in device fabrication [5]. The construction and development of devices incorporating the main components of natural biological photosynthesis, mainly the solar-to-electrical energy converting protein-pigment complexes, has emerged as an interesting area of research over the past few years. These bio-sensitized devices could be used for potential future applications such as photo-switchable biosensors, solar-to-chemical, and solar-to-electrical energy converting systems and establish a basis towards a renewable energy economy [6,7]. 

Two protein–pigment complexes that have shown great promise in this area of biologically-based electronic devices are bacteriorhodopsin (bR) and photosystem I (PSI). PSI is one of the primary reaction centers that are central to oxygenic photosynthesis and is comprised of approximately 12–14 protein subunits, 100 chlorophyll molecules, two phylloquinones, three [4Fe-4S] clusters and 20 carotenoids. Furthermore, PSI is able to form higher oligomeric states in many different photosynthetic organisms, reaching sizes of 1.4 MDa, making it among the most complex protein complexes found in nature [8,9,10,11]. In vivo, PSI acts as a photoactivated cytochrome *c*:ferredoxin oxidoreductase, catalyzing a light dependent unidirectional electron transfer across the thylakoid membrane [12]. bR is a membrane-bound protein–pigment complex found in archaebacteria such as *Halobacterium salinarum* and is comprised of a single 27 kDa subunit and a single Schiff base linked retinal pigment. Intriguingly, bR forms a trimeric structure in vivo similar to PSI [13]. The three-dimensional structure of both proteins and locations of their pigment moieties are shown in Figure 1, and Table 1 summarizes the structural and optical properties of both, as well as the organismal source of both proteins used in this study. However, bR uses photoexcitation to catalyze the transfer of protons across the membrane it is embedded into [14]. Both protein–pigment complexes have shown remarkable thermal stability and their structural, biochemical, and photophysical properties have been extensively studied [15,16]. Similarly, both PSI and bR have also been widely used in a variety of biotechnological applications, most notably in BSSCs [17,18,19,20,21,22,23,24,25,26,27,28,29,30,31,32,33,34,35,36,37,38,39].

BSSCs consist of a biological pigment immobilized on a semiconductor photoanode, connected to a cathode through a redox system solution and a supporting electrolyte, as illustrated in Figure 2A [5]. The conventional architecture of a BSSC consists of five primary components (Figure 2): (*i*) a photosensitive biological dye, (*ii*) a transparent photoanode, (*iii*) a semiconductor coating on the photoanode, (*iv*) a redox electrolyte for dye regeneration, and finally, (*v*) a counter electrode [16,26,40,41,42]. After photonic excitation, the dye molecules are excited from their ground state to a higher energy state, and the electrons are promoted from the HOMO orbital to the LUMO orbital, generating electron–hole pairs [41,42]. Then, effective charge separation is achieved by the oxidation of the excited sensitizer molecule, and the generated electron is injected to the conduction band of the semiconductor on the photoanode, and the hole remains behind in the oxidized dye molecule [7]. Then, the electrons diffuse through the semiconductor to the photoanode substrate, where they travel through an external circuit and perform work before arriving at the cathode, generating the current. During that time, the redox mediators in the electrolyte regenerate the sensitizer molecule [43,44]. In return, the oxidized redox mediator diffuses to the counter electrode (CE) and gets regenerated [41]. Within this basic operational outline, there is great flexibility in the selection and design of materials, along with many different interfaces to optimize for device improvement.

In BSSCs, the CE catalyzes the reduction of the electrolyte after electron injection. The CE is expected to be both highly conductive and highly catalytic. Large surface areas and small particle sizes are preferred to generate more active sites and enhance the electrocatalytic activity of the CE [52,53]. In most cases, platinum-coated conductive glass has been used for CEs due to its high conductivity and catalytic activity. However, Pt is expensive and is susceptible to corrosion and irreversible redox reactions [54], making it a poor material choice for incorporation into some sustainable devices, especially BSSCs, which strive to utilize more abundant resources for CE materials. Different forms of carbon allotropes, inorganic materials, and conductive polymers are emerging as promising CEs, and among the most promising are poly (3,4-ethylene dioxythiophene) (PEDOT) and alternative carbon nanomaterials [55,56,57,58,59,60,61,62]. PEDOT is a conductive polymer with desirable physical properties including optical transparency, smooth surface morphology, solution processability, light weight, low cost, electrochemical stability, mechanical flexibility, and a high work function [63]. However, due to the acidity and extremely hygroscopic nature of PEDOT films, it is often modified during the electrodeposition process to incorporate additives that tune its properties for specific systems and enhance its performance as a CE in BSSCs. These include carbon nanostructures such as carbon nanotubes (CNTs), which have many advantages such as high surface area, exceptional charge carrier mobility, and remarkable mechanical strength and stability, beyond the abundance of carbon on Earth [64]. Considering the many beneficial physical properties of PEDOT and CNTs, it can be predicted that the composites of PEDOT and carbon nanostructures may be more efficient in enhancing the performance of BSSCs compared to other materials that have already been reported for transparent charge transport layer applications [65]. Other commonly used carbon allotropes for CEs include graphene-based systems. Pi-system modified graphene-based CEs have already been reported on for use in PSI-sensitized solar cells with photocurrent densities of 135 µA/cm^2^ [66]. Graphene has excellent conductivity compared to certain other carbon allotropes and is highly abundant, both of which are desirable characteristics for sustainable BSSC designs [67].

In BSSCs, the electrolyte is responsible for reduction of the photosensitizer back to its ground state and selection of a proper electrolyte and redox mediator are crucial to device performance and longevity. As such, careful selection of the redox mediator is crucial for the alignment of its midpoint potential *E_m_* for donation to the HOMO of the dye after photoexcitation, as depicted in step 4 in Figure 2B. This allows for the minimization of overpotential losses in BSSCs. BSSCs have historically used the canonical I^−^/I_3_^−^ redox mediator pair, yet this system has multiple drawbacks including high corrosivity and toxicity, generation of radical species, significant absorption in the same region of the UV-visible spectrum as biological photosensitizers, and an unfavorable midpoint potential (*E*_m_) as compared to the HOMO of many biological materials giving low driving potentials for improved electron transfer rates [68]. Further, the iodide/tri-iodide pair is also commonly dissolved in organic solvents which are undesirable in BSSCs due to their high volatility, toxicity, and/or explosive nature of many of these solvents, along with water leakage into the device either during fabrication or while being used [69]. As such, the development of aqueous, biocompatible electrolytes and redox mediators is key. Most work in the field of DSSCs has focused on the use of cobalt-based redox mediators to try and replace the I^−^/I_3_^−^ pair [58,70,71,72,73,74,75] due to the high abundance of cobalt and its stable, reversible redox properties. Previous work from our group has described the synthesis and characterization of novel aqueous-soluble cobalt-based redox mediators and their ability to directly reduce PSI in vitro and the fabrication of fully aqueous BSSCs using the same mediator [44].

In this study, we describe a facile technique of fabrication of a CE based o a highly conductive composite of PEDOT and multi-walled carbon nanotubes (MWCNTs), and the use of these PEDOT/CNT counter electrodes with an aqueous bipyridine-based cobalt redox mediator to fabricate BSSCs using both PSI and bR. A schematic of the energy levels and device design is shown in Figure 2A,B, along with an overlaid visible absorbance spectrum of these complexes (Figure 2C) showcasing the potential for devices with multiple biological photosensitizers to utilize a greater optical cross-section for photocurrent generation. The compatibility of this novel counter electrode and redox mediator scheme with more than one biological photosensitizer is remarkable and may serve as a platform for developing more broadly biocompatible BSSC components to improve the biological-inorganic interface.

## 2. Results

### 2.1. Counter Electrode Characterization

We analyzed the performance and characteristics of three different electrode materials, PEDOT/CNT, platinum nanoparticles (Pt NPs), and graphene, prepared on FTO glass substrates. Then we measured the performance of the BSSC using these substrates as counter electrodes. The morphology and surface characteristics of the photoanodes and all counter electrodes used in this study were analyzed using scanning electron microscopy (SEM). The resulting micrographs can be seen in Figure 3. In Figure 3A, the surface topography of the PEDOT/CNT counter electrode can be observed in this top-down image of the electrode face. The spherical structures, ca. 400–500 nm in size, are likely the PEDOT polymer, while the small protrusions are where the doped carbon nanotubes have integrated into the polymeric structure during electropolymerization. The rough surface area of the spherical polymer particles may allow for an improved contact to reduce the mediator in the electrolyte.

In Figure 3B, a cross-section view of the sintered TiO_2_ layer on the photoanode is depicted showing the even thickness of the deposited porous semiconducting layer. The thickness of the TiO_2_ layer on the FTO coated glass is 40 µm. The highly porous three-dimensional nature of the sintered semiconductor TiO_2_ layer is especially notable in the top-down view of the electrode surface in Figure 3C. This allows for improved adsorption of photosensitizers such as PSI and bR. The general morphology aspects of all counter electrodes and PSI deposited on the TiO_2_-coated photoanode can be seen in Figure 4A.

The ability of PEDOT/CNT, graphene, and Pt NPs to interact constructively with our aqueous-soluble Co^II/III^ redox mediator was then tested by cyclic voltammetry, using the variable counter electrodes as the working electrode in an analytical electrochemical setup. The resultant voltammograms are shown in Figure 4B, with glassy carbon and Pt wire used as standard working electrodes for comparison. Each voltammogram was normalized for peak height to take into account variable working electrode area. The specific redox reaction measured in Figure 4B is described in Equation (1) and Figure 5 below [44,76], where R = tris(4,4′-di-methoxy-2,2′-bipyridine).
(1)$$[\mathrm{Co}{\left(\mathrm{bpy}-R_{2}\right)}_{3}{\mathrm{Cl}}_{2\;}]{\;}^{3+}+{\;e}^{-}⇌[\mathrm{Co}{\left(\mathrm{bpy}-R_{2}\right)}_{3}{\mathrm{Cl}}_{2\;}]{\;}^{2+}$$

The oxidation potential *E*_ox_, reduction potential *E*_red_, and midpoint potential *E*_m_, of the Co^II/III^ redox mediator were measured using each electrode and the obtained values are reported in Table 2. The aqueous-soluble Co^II/III^ redox mediator was able to interact electrochemically with each eletrode material, displaying quasi-reversibility. The *E*_m_ measured (−0.045–0.236 V vs. NHE) was higher than the HOMO of both PSI and bR with all electrodes tested. The *E*_m_ of the mediator being more negative than the HOMO of bR [48] and of PSI [47] indicates that electron transfer from the Co^II/III^ redox mediator to either PSI or bR is thermodynamically favorable, and a closer *E*_m_ value to those of the protein–pigment complexes represents lower overpotential losses [75].

The varying counter electrodes were next used for functional comparison by incorporation into PSI-sensitized solar cells (PSI-SSCs) and testing for photovoltaic performance and output. The resulting current density–voltage (*J*–V) and photochronoamperometry curves can be seen in Figure 6 below.

The photovoltaic parameters of the three devices, short-circuit current (*J_SC_*), open-circuit voltage (*V_OC_*), fill factor (FF), and power conversion efficiency (PCE) are summarized in Table 3. The greatest open-circuit *V_OC_*, *J_SC_*, and PCE were obtained with the PEDOT/CNT-based PSI-SSC, which outperformed both the graphene and Pt NPs counter electrodes. The *V_OC_*, *J_SC_*, and PCE of the PEDOT/CNT cell were −132 mV, 10.0 µA/cm^2^**,** and 0.33%, respectively. The photocurrent density response of the PEDOT/CNT PSI-SSC was nearly five times larger than that of Pt NPs’ (2.22 μA/cm^2^) and eight times greater than graphene’s (1.3 μA/cm^2^). The photocurrent response of the PSI-SSCs at the *V_OC_* is plotted in Figure 6A. All of the PSI-SSCs exhibited a current increase in response to illumination, with PEDOT/CNT and Pt NPs giving similar photocurrent and higher than that of the graphene counter electrode.

### 2.2. Comparison of Liquid and Gel Electrolyte Composition on PSI-SSC Output and Efficiency

Once the PEDOT/CNT counter electrode was determined to be the best performing with the aqueous Co^II/III^ mediator and PSI as a photosensitizer, the electrolyte was next varied to compare gel vs. liquid consistencies for BSSC device performance. Previous studies on gel-based electrolytes have reported on their abilities to overcome some of the largest issues relevant to liquid electrolyte-containing devices, such as sealant failure, volatile solvent evaporation, greater electron recombination, and device leakage over time [77,78]. In Figure 7, a gel-based electrolyte system was compared to the liquid electrolyte to compare their performances with the PEDOT/CNT counter electrode in PSI-SSCs. *J*–V curves comparing gel and liquid electrolytes can be seen in Figure 7A, with the extracted photovoltaic parameters reported in Table 4. The *V_OC_* of the gel electrolyte-based device was improved by −60 mV relative to the liquid electrolyte device. The gel and liquid electrolyte devices had comparable fill factors of 26 and 25%, and a *J_SC_* of 9.67 and 9.94 µA/cm^2^, respectively. However, the gel electrolyte device had an improved PCE of 0.48%, compared to 0.33% for the liquid electrolyte device. Based on the improved *V_OC_* and PCE of the gel electrolyte device, the gel-based system was able to reduce resistance losses and improve device performance. The *J_SC_* is unchanged as light absorption and transduction into current should not affected by the electrolyte composition.

Photochoronamperometry experiments were next performed to measure the photocurrent as a response to illumination and the results are shown in Figure 6B. Both liquid- and gel electrolyte-based devices showed improved current as illumination was increased, as expected. The gel electrolyte-based device generated a larger photocurrent than the liquid-based device, and even upon stabilization of the photocurrent the gel-based device had comparable current density output to the liquid-based device under 50% less irradiance. Taken together, the gel electrolyte with PEDOT/CNT counter electrode-based device has the best performance of all systems tested in this study for PSI-sensitized BSSCs.

### 2.3. Aqueous Co^II/III^ Redox Mediator Gel Electrolytes and PEDOT/CNT Counter Electrodes Are Compatible with Multiple Proteins

Once an improved counter electrode and electrolyte system for PSI-SSCs was identified, the suitability of this novel device fabrication scheme for other biological sensitizers was tested by the utilization of the protein bacteriorhodopsin (bR) instead of PSI. While PSI essentially behaves as a biological diode, in vivo Br performs activity as a light-activated proton pump. Similarly, while both protein–pigment complexes are membrane-bound and are typically purified using detergents, they have significantly different buffer requirements for photoactivity retainment and stability of their protein–pigment complex [79,80,81,82,83,84]. This is generally true for any biological components that may be incorporated in BSSCs, and to date, the improvement of biocompatibility remains a research area of great interest.

To test how generally biocompatible our electrolyte and counter electrode scheme was, a series of BSSCs were fabricated that utilized the same gel-based electrolyte using the aqueous Co^II/III^ redox mediator and PEDOT/CNT counter electrodes, but were sensitized with either PSI, bR, or a mixture of both proteins. For these devices, the *J*–V curves and photocurrent density under different illumination regimes were measured, due to the different absorption spectra of the two protein–pigment complexes (Figure 2C). The irradiance spectra from the illumination source with different filters is shown in Figure 8A. The photochronoamperometry experiments showing light response are shown below in Figure 8B,C. The calculated *V_OC_*, *J_SC_*, FF, and PCE for each cell based on the *J*–V curves are summarized in Table 5 below. The bR-SSC had the best overall performance, with a *V_OC_* nearly 40 mV more negative than the PSI-SSC, and nearly 2.4 more µA/cm^2^ photocurrent density. The PCE was also markedly improved in bR-SSC by approximately 0.3% compared to the PSI-SSC.

The photocurrent response to varying light regimes for differing photoactivation of the devices was next tested, as the two protein–pigment complexes are optimally excited by differing wavelengths of light. For the PSI-SSC, the photocurrent density reached ~1 µA/cm^2^ under green illumination (~525 nm) and ~0.15 µA/cm^2^ under red light illumination (~620 nm). The bR-SSC had similar photocurrent densities under the green light illumination, despite its improved photoactivation. PSI likely still had significant photocurrent compared to the blank cell due to its greater number of pigment per PSI monomer (~100) [8] than bR (1) allowing for more efficient light harvesting. As PSI absorbs lower energy wavelengths more readily than bR, the PSI-SSC yielded an improved photocurrent density as compared to the bR-SSC under red light illumination (~620 nm) (Figure 8C). Both sensitized cells yielded improved photocurrent densities upon illumination as compared to the blanks, suggesting that the PEDOT/CNT counter electrode and aqueous gel Co^II/III^ electrolyte fabrication scheme is compatible with both membrane protein–pigment complexes and allows for direct reduction of bR by the bipyridine-based mediator we have previously reported on for PSI-SSCs [44].

The differing photocurrent response to illumination on either the anodic or cathodic side of the PSI- and bR-sensitized devices was also tested, shown in Figure 9A. The measured photocurrent densities generated were over 10-fold greater upon illumination through the back of the anodic surface for both the PSI- and bR-SSCs, and consistent over multiple illuminations in the same photochronoamperometry trace. This may be due to the highly scattering nature of our regular TiO_2_ semiconductor layer and the relatively high UV-visible light absorption of the PEDOT/CNT CE and the electrolyte.

The stability of the PSI- and bR- sensitized devices were tested over a period of two weeks to assess the longevity of the devices. The devices were illuminated for a total of 5 min for 3 total repetitions each day over 15 days. The average photocurrent density obtained each day along with standard deviation is plotted in Figure 9B. The average photocurrent density trace is reported for days 1 and 14 in Figure 9C for both the PSI-SSC and bR-SSC tested. Interestingly, both the bR- and PSI-SSCs showed robust photocurrent densities over the entire two weeks of testing with no significant losses. The bR-SSC had an average photocurrent density of ~1.7 µA/cm^2^ and the PSI-SSC had an average photocurrent density of ~3.6 µA/cm^2^ for the entire tested period. The compatibility of the PEDOT/CNT cathode and aqueous gel-based electrolyte with the aqueous Co^II/III^ redox mediator with the two protein–pigment complexes is further proven here by the performance stability tests of both devices.

## 3. Discussion

Biologically-derived light harvesting components for photovoltaic applications have commercial potential due to the wide availability and ease of directed growth of different organisms, more environmentally friendly production, and lower cost of device fabrication. One such technology is the fabrication of sensitized solar cells based on biological proteins as sensitizers, also known as bio-sensitized solar cells (BSSCs). However, BSSC technology faces many challenges to become commercially competitive, including the enhancement of photocurrent generation, improvement of conversion efficiencies, flexibility, scaling, and long-term stability [58,85,86,87]. Many BSSC studies only report the photocurrent of the cells studied, and do not include more detailed studies on conversion efficiencies, and rarely demonstrate stability over time. One other significant factor in BSSC development that is often not carefully considered in device fabrication schemes is the stability of the biological component. In this report, we describe a systematic testing of multiple counter electrodes and compare a liquid-based vs a gel-based electrolyte system that is compatible with two unique biological protein–pigment complex photosensitizers, PSI and bR. The best photocurrent and efficiency results obtained in our studies were from PSI- and bR-sensitized solar cells using a device design incorporating an aqueous gel-based electrolyte with a Co^II/III^ redox mediator and a PEDOT/CNT counter electrode.

We have previously reported the fabrication of PEDOT/CNT CEs for biosensors [88,89], yet this is the first report of their use in BSSCs and with cobalt-based redox mediators. Our PEDOT/CNT CEs were similar in morphology to those manufactured in previous studies from our group, with all samples exhibiting continuous, well-coated electrode surfaces. Granular morphologies of ca. 500 nm diameter particles could be seen, comparable to results reported by other groups for PEDOT films [90]. The obtained rough surface pattern is likely a result from the micellar system used for electrochemical deposition allowing for increased surface area for the redox mediator to interact. The results of the improved output and efficiencies of SSCs using PEDOT/CNT CEs with our aqueous bipyridine-based Co^II/III^ redox mediator is consistent with other published studies, where it has been reported that a PEDOT-based CE was able to generate greater FF and PCE than Pt based electrodes [91]. This was attributed to the ability of PEDOT/CNT films to more effectively suppress carrier recombination and promote carrier extraction, improving photocurrent generation in the device [92], and the ability to mitigate the diffusional limitation of Co species which has been reported to affect similar chemical species [76]. Previous groups have reported on the incorporation of PEDOT-based CEs with cobalt-based redox mediators in inorganic DSSCs with similar results [93]. This improvement of the electrochemical response of PEDOT/CNT CEs has been attributed to their excellent electrical conductivity [64,89], intrinsic electrocatalytic activity [94], and the high heterogeneous electron transfer rates of the embedded MWCNT, where unique hollow structure and edge-plane-like defects of MWCNTs, such as open ends, enhance these characteristics [95]. This improved conductivity can also contribute to the restriction of recombination events, prolonging the lifetime of active species in the electrolyte [96].

Further, the work function of MWCNTs (~4.5–5.1 eV) is closer to that of PEDOT (~5.0 eV), which may aid in the reduction of overpotential losses and improved device output and efficiency [96,97]. Our bipyridine-based Co^II/III^ redox mediator had an *E*_m_ closest to the HOMO of both biological protein–pigment complex used when measured using the PEDOT/CNT electrode as the working electrode, likely aiding in further reducing overpotential losses in the devices. On the other hand, the platinum work function is located at a more positive potential vs NHE (0.66–1.46 V vs. NHE) [98] than the *E*_m_ of the Co^II/III^ redox mediator (0.05–0.34 V vs. NHE), therefore the energy alignment is not thermodynamically favorable for catalyzing this electron transfer event [99]. In the case of graphene fabricated by CVD, it mostly interacts with the electrolyte through the basal plane, which has the lowest heterogeneous electron transfer constant of the different carbon allotropes and displays a very poor electrocatalytic activity for the mediator redox reaction [100]. Overall, the reduction of the Co^II/III^ redox mediator devices with PEDOT/CNT counter electrodes is taking place with reduced energy losses than with the Pt NPs and graphene, contributing to its overall improved photovoltaic performance in the BSSCs studied.

Dye-sensitized solar cells have traditionally utilized liquid electrolytes and have achieved some of the greatest efficiencies reported with them [101]. While liquid electrolytes have certain desirable characteristics such as improved wettability of the cell, redox mediator solubility, and greater electrode surface contact, there are many practical issues such as leakage, desorption of the sensitized dye (or protein), and toxicity of many of the most common liquid electrolytes used, along with increased rates of recombination events leading to decreased amounts of electrons available to reduce the photosensitizer [18,43,77,78,82,102]. Gel- and solid-based conductive electrolytes have been an area of much interest to attempt to address these deficiencies and have also been shown to aid in the reduction of charge recombination at TiO_2_/dye/electrolyte interfaces [16,87,103] and to improve the mechanical strength of the fabricated cells. In the results of this study, we found that utilization of a PEG-based gel electrolyte yielded a greater *V_OC_* and PCE than a liquid-based system. We also report that this electrolyte system is stable for at least 15 days with no losses of photocurrent density response upon device illumination. A similar gel-based electrolyte system has been reported to provide better interaction of the photosensitizer with the redox-active components of the electrolyte system, which may explain the improved longevity we have reported for our bR- and PSI-SSCs.

The comparative performance of the bR- and PSI-SSCs was also evaluated as a function of irradiance spectra, as both proteins have preferential wavelengths for photoexcitation and differing quantities of pigments for light harvesting. Both sensitized devices performed better than the blank cell, which is due to the incorporation of the biological photosensitizer and proving their constructive photoactivity in the device. Both devices yielded similar photocurrent density response under green light illumination, yet unsurprisingly, the PSI-SSC outperformed the bR-SSC under red light illumination as it is more efficient at utilization of lower-energy photons for photocurrent generation. Interestingly, combining both proteins in the same device through simple drop casting yielded a device with reduced performance, likely due to the need for more directed orientation for forward electron injection through the device. The difference in photocurrent density output between the two photosynthetic proteins may be explained by a variety of factors such as the improved pigment density per unit of photosystem I as compared to bR or perhaps the function of bR as a proton pump is adding another layer of complexity to the system behavior by affecting the local environment near the TiO_2_ semiconductor layer.

Once the performance of the BSSCs sensitized with either protein was analyzed, the long-term photocurrent stability of fabricated devices was investigated. The operational stability was assessed by monitoring photocurrent response over the course of 15 days under the same repeated illumination scheme for multiple traces each day. No loss in photocurrent density was noted over the tested period, suggesting that device longevity is likely longer than this period tested as well. This indicates the viability of both PSI- and bR-based quasi-solid state devices for real world settings where lighting will not always consistent and go through light–dark cycles, and that the photoactivity of the cell is able to be stabilized and retained for improved longevity. PSI-SSCs had generally larger photocurrent density generation in our studies than bR-SSCs, likely due to the improved pigment–protein ratio of PSI, the greater UV-Visible absorption of chlorophyll pigments compared to retinal, and the terminal electron transfer cofactor in PSI being closer to the protein’s surface as compared to the buried retinal of bR being more insulated by its protein environment. However, upon analysis of the *J*–V curves of the two gel-based BSSCs, bR had a calculated *J_SC_* photocurrent density of 13.67 µA/cm^2^, slightly outperforming the PSI-sensitized device. The bR device also exhibited a PCE 48% greater than the PSI-sensitized device, 1.04%, as compared to 0.7%.

Taken together, these results effectively demonstrate the robustness of the designed BSSC TiO_2_(bR/PSI)/PEDOT/CNT device fabrication scheme and demonstrates its compatibility with multiple biological protein–pigment sensitizers in a manner that preserves their photoactivity. The development of more compatible device designs, and ideally more universally biocompatible device designs is critical for further improvements of bio-sensitized solar cells and is hopefully an area of increased research focus in future studies. This work, and others in the field, help to showcase the importance of further development of bio-sensitized photovoltaics from low-cost and sustainable materials to help meet growing energy demands worldwide.

## 4. Materials and Methods

### 4.1. Counter Electrode Fabrication

Platinized counter electrodes were fabricated by doctor blading Platisol T/SP from Solaronix (ref. no. 41211) onto conductive FTO glass electrodes (~25 × 25 × 2 mm) that were masked off using Scotch tape to define active electrode area. The FTO coated unpolished float glass was from Delta Technologies (Loveland, CO, USA) and had Rs = 5–10 ohm. Tape was then removed and platinized counter electrodes were then sintered starting in a cold furnace up to 450 °C where they were held for 1 h. Graphene counter electrodes (~25 × 25 × 2 mm) were a gift from General Graphene (Knoxville, TN, USA) with a four layer-thick deposition of graphene on FTO glass. The PMMA coating on the graphene electrodes was removed by washing with acetone, followed by intensive rinsing with MilliQ H_2_O. The multiwall carbon nanotubes (MWCNTs) were purchased from CheapTubes.com (SKU #030106). PEDOT/CNT electrodes were electropolymerized from water dispersion using SDS as surfactant at the CMC (CMC = 8.2 mM) and 0.35% mass concentration of MWCNTs. SDS/CNT emulsions were sonicated before and after the addition of the monomer, EDOT (10 mM). The dispersion was deposited on the electrode surface using galvanostatic conditions on an Autolab Potentiostat. Conductive glass electrode was used as working electrode, platinum foil as counter electrode, and Ag|AgCl (KCl 3.0 M) was used as a reference electrode. The electrical polymerization was carried out with a current density of 1 mA/cm^2^ using a potential limit of 1.9 V for 240 s (ca. 120 mC/cm^2^ of charge density). Following the PEDOT:SDS:MWCNT deposition, the electrodes were intensively rinsed with deionized water.

### 4.2. Protein Isolation

Bacteriorhodopsin protein was obtained from Dr. Renugopalakrishnan. It was purified from *H. salinarum* as described in [31]. In brief, *H. salinarum* frozen cells were resuspended in 250 mL of basal salt without peptone. The cells were then dialyzed overnight in 2 L of 0.1 M NaCl to lyse the cells. The lysate was centrifuged and the membrane pellet was resuspended and Dounce homogenized, and then centrifuged again. The membrane pellet was then separated using sucrose density gradient ultracentrifugation and the lower purple band was harvested. This purple membrane was then solubilized using Triton X-100 detergent followed by gel filtration in deoxycholate solution to yield pure delipidated bR protein. The protein was then lyophilized and then resuspended in 20 mM HEPES, 200 mM KCl, 0.06% Triton X-100 pH 8.0 prior to use.

PSI trimer was isolated and purified from *T. elongatus* as previously reported [104]. Briefly, *T. elongatus* frozen cells were resuspended in wash buffer (25 mM MES, 20 mM MgCl_2_ pH 6.4) and Dounce homogenized. Lysozyme was added and the suspension was incubated to allow for cell wall degradation. The suspension was pelleted and washed with fresh wash buffer before being passed twice through a French press. The lysate was centrifuged and the pelleted membrane fragments were washed again. *N*-dodecyl β-d-maltoside (β-DDM) was added to the resuspended pellet to 0.4%, which was then incubated for 1 h at 37 °C. The insoluble material was removed via centrifugation and then the solubilized material from the membrane pellet was separated using sucrose density gradient ultracentrifugation for 14 h, after which the lowest band containing trimeric PSI was harvested. The harvested PSI was then purified using HPLC before aliquoting and storage.

### 4.3. Cobalt Redox Mediator Synthesis

Aqueous bipyridine-based cobalt redox mediators were synthesized as described previously in [44]. Briefly, bipyridine cobalt complexes bearing methoxy functional groups were synthesized using chloride as counterion. A total of 3 molar equivalents of 4,4′-dimethoxy-2,2′-bipyridine was reacted with 1 molar equivalent of cobalt (II) chloride hexahydrate in methanol for 3–4 h at 55 °C. Afterwards, the methanol was evaporated under reduced pressure and the resulting solid was washed with cold diethyl ether to yield the chloride salts in quantitative yield. Bi-pyridyl (*tert*-butyl and methoxy) ligands were purchased from Sigma-Aldrich and used without purification. The preparation was performed following protocols in the literature [76,105,106].

### 4.4. Cyclic Voltammetry of Co Redox Mediator with Variable Electrodes

All working electrodes were rinsed intensively with di H_2_O prior to testing, and Pt and glassy carbon (GC) were also polished prior to rinsing. All cyclic voltammograms were ran at a slew rate of 100 mV/s. The Co redox mediator was present at a concentration of 2 mM in an aqueous buffer of 20 mM HEPES and 200 mM KCl as a supporting electrolyte with pH 8.0. The area of the various working electrodes tested was highly variable and all resultant voltammograms had peak heights normalized to −1 or +1 to allow for easier comparison of peak shifts and separation.

### 4.5. Scanning Electron Microscopy

Electrode samples were prepared as described previously. A Zeiss Auriga Crossbeam FIB/SEM was used for all scanning microscopy images. All counter electrodes were imaged at EHT = 5 kV and TiO_2_ coated electrodes were imaged at EHT = 2 kV to improve image quality.

### 4.6. Device Fabrication

A TiO_2_ suspension comprised of 0.8 g anatase TiO_2_ (9–12 nm particle size), 1.2 mL 0.1 M HNO_3_, 0.024 g polyethylene glycol 8000, and 0.06 mL Triton X-100 (Anatrace, Maumee, OH, USA) was sonicated for 1 h and then stirred overnight. Conductive FTO glass electrodes (~25 × 25 × 2 mm) were masked off using Scotch tape to denote the active electrode area and then the TiO_2_ suspension was doctor bladed on. The electrodes were air-dried, the tape was removed, and then the electrodes were sintered at 450 °C for 1 h to the bare photoanodes. Photoanodes were stored in Dri-Rite until use. Cured polydimethylsiloxane (PDMS) was used to create a ~1 cm^2^ active area for drop coating photoanodes. The precise surface area was calculated using ImageJ (Ver. 1.53). Protein-containing devices were drop coated with 20 µL of each respective protein (PSI at ~12 µM, bR at ~50 µM due to different light harvesting abilities). The 2 electrodes were then offset and mechanically compressed together with the PDMS spacer in between, with dual clamps on opposing sides holding the device together, and the electrolyte was introduced using the 0.5 mm drilled holes in the CE active area. Then, the holes were sealed with a PDMS patch on top. The liquid electrolyte for PSI devices consisted of 0.03 M Co^II/III^ mediator, 50 mM MES, 0.03% β-DDM at pH 6.4. The liquid electrolyte for bR devices consisted of 0.03 M Co^II/III^ mediator, 20 mM Tris, 200 mM KCl at pH 8.0. All gel electrolytes were the same composition as the liquid electrolyte with the addition of 25% *w*/*v* PEG 10,000 (Sigma CAS# 25322-68-3).

### 4.7. Device Testing

Devices were allowed to equilibrate for wetting and electrolyte integration for 30 min prior to device testing and measurements. Device illumination was performed using a Schott KL-2500 LED light source with inset filter holder and Schott red light filter (MOS-258-303) for all red actinic light experiments, and a Knight Optical dichroic bandpass filter 525 nm (#525FDC25, Maidstone, UK) for all green actinic light experiments. All photochronoamperometric measurements were taken using a Bio-Logic SP-50 potentiostat and EC-Lab software for data collection. Data plotting and analysis was done in Origin Pro 2019, QSOAS [107], and GraphPad Prism (Ver. 9.3.1).

## Figures and Tables

**Figure 1 ijms-23-03865-f001:**
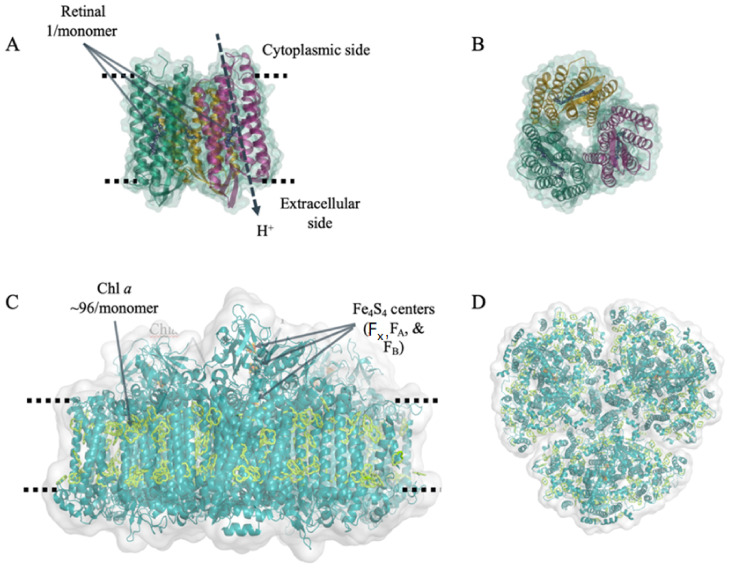
Structure of bacteriorhodopsin and photosystem I. Three-dimensional structure of pigment-protein complexes: (**A**) bacteriorhodopsin side view and (**B**) top view. (**C**) Photosystem I side view and (**D**) top view. The three subunits of bR are shown in orange, purple, and green respectively, with retinal shown in blue. In PSI, all of the protein is shown in teal with the chlorophyll light-harvesting antenna shown in green. For both proteins, the dashed line represents the position of the membrane each complex is embedded in vivo.

**Figure 2 ijms-23-03865-f002:**
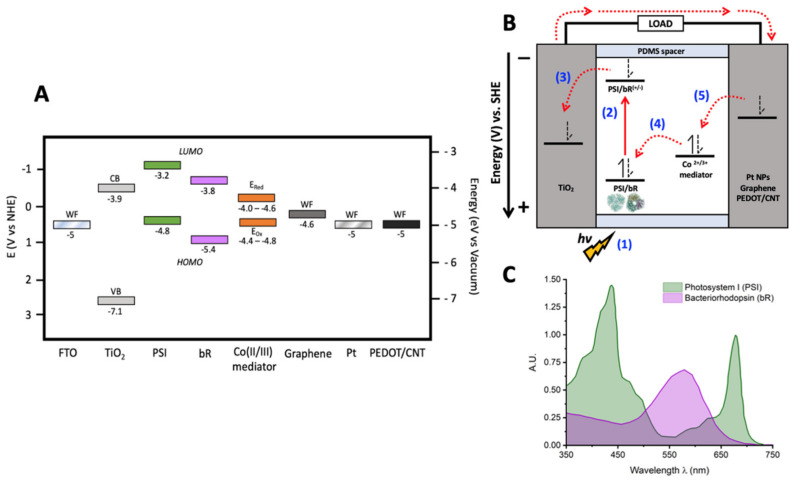
Schematics of PSI or bR sensitized BSSCs. (**A**) Energy levels diagram of all device components used in this study. Ranges for the values of the Co (II/III) mediator come from the data reported later in this study. Other values and conversions used come from references [45,46,47,48,49,50,51]. WF, work function; CB, conduction band; VB, valence band; LUMO, lowest unoccupied molecular orbital; HOMO, highest occupied molecular orbital. (**B**) Fundamental processes and constituent components of a bio-sensitized solar cell (BSSC). (**C**) UV-vis absorbance spectra of PSI and bR.

**Figure 3 ijms-23-03865-f003:**
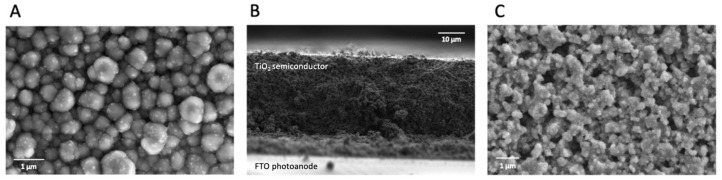
Surface characteristics of photoanodes and counter electrodes. SEM micrographs of (**A**) PEDOT/CNT (**B**) cross-section view and (**C**) top view of sintered TiO_2_ all on FTO.

**Figure 4 ijms-23-03865-f004:**
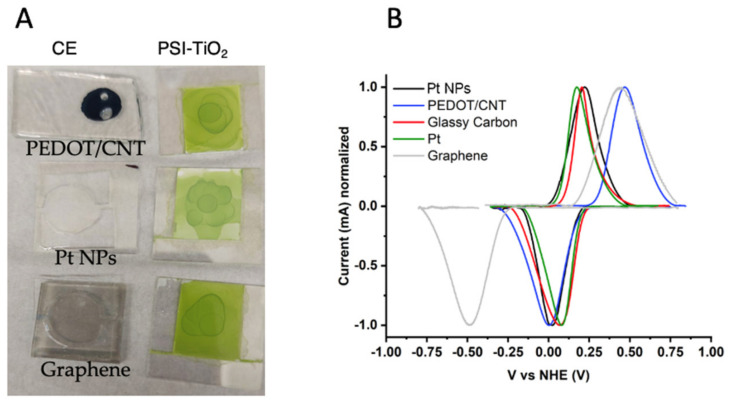
Counter electrode characterization. (**A**) Photograph of the three PSI-SSC with different counter electrodes (left) and PSI-sensitized TiO_2_ photoanodes (right) on FTO glass. (**B**) Cyclic voltammograms of the of aqueous-soluble bipyridine cobalt (II/III) redox mediator using different working electrodes. All voltammograms were measured at 100 mV/s, baseline corrected, and current was normalized to account for variation in working electrode surface area. Pt wire and a saturated calomel electrode were used in all measurements as the counter and reference electrodes, respectively.

**Figure 5 ijms-23-03865-f005:**
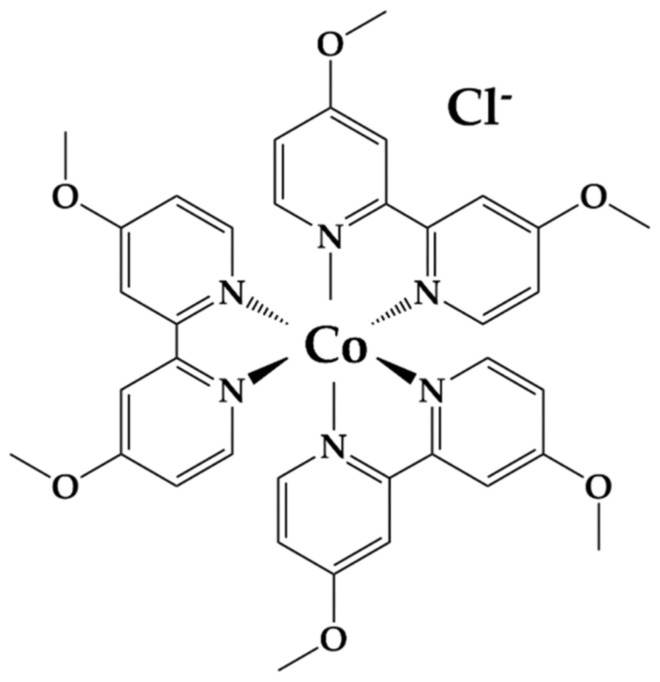
Chemical structure of aqueous-soluble Co^II/III^ redox mediator.

**Figure 6 ijms-23-03865-f006:**
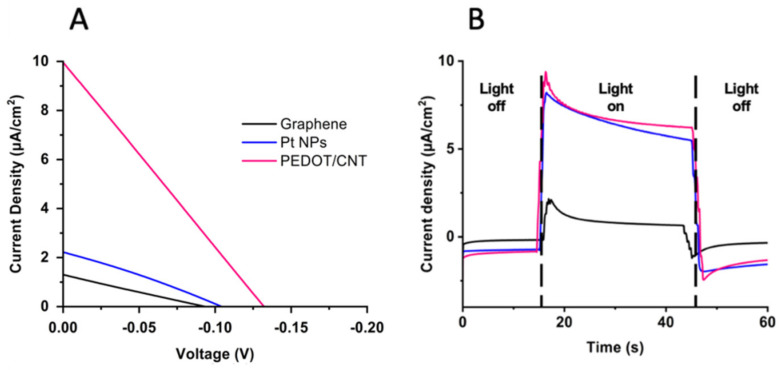
Design and performance of PSI-sensitized solar cells (PSI--SSC) with variable counter electrodes. (**A**) White light illuminated current density-voltage (*J*V) curves of the three PSI-SSCs. (**B**) Representative chronoamperometry curves of the three PSI-SSCs illuminated with white light.

**Figure 7 ijms-23-03865-f007:**
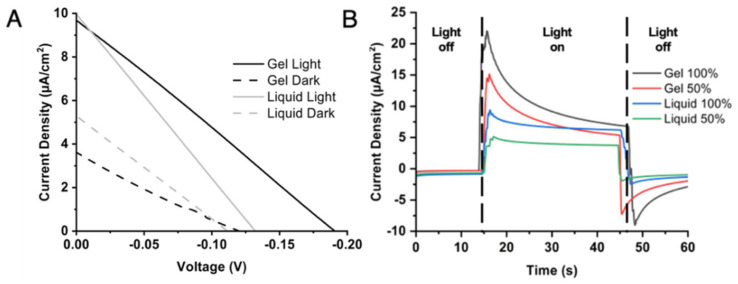
Electrochemical analysis and performance of PSI-sensitized solar cells (PSI-SSC) using liquid and gel Co^II/III^-based electrolytes. (**A**) Dark and white light illuminated *J*-V curves of the fabricated devices using liquid and gel electrolytes. (**B**) Representative chronoamperometry experiments measured at different white light illumination intensities.

**Figure 8 ijms-23-03865-f008:**
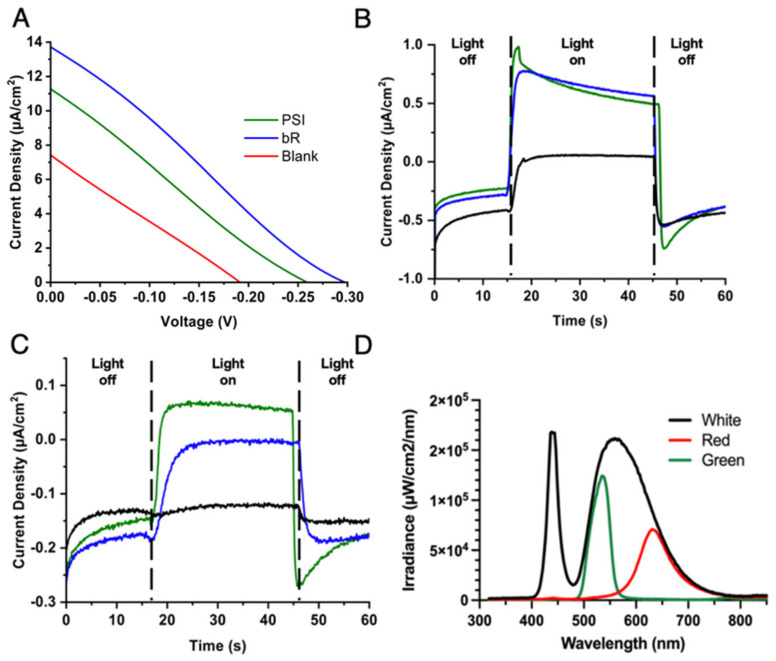
PEDOT/CNT counter electrodes and a gel-based Co^II/III^ electrolyte are also compatible with bR-SSCs. (**A**) *J*-V curves of different BSSCs illuminated under white irradiance at 100% lamp intensity output. Chronoamperometry measured at 0 V under (**B**) 50% green light and (**C**) 50% red light illumination. Black lines in B and C are blank cells, with green lines showing PSI-SSC data and blue lines showing bR-SSC data. (**D**) Plot showing the spectral irradiance of the LED illumination source at 10% intensity.

**Figure 9 ijms-23-03865-f009:**
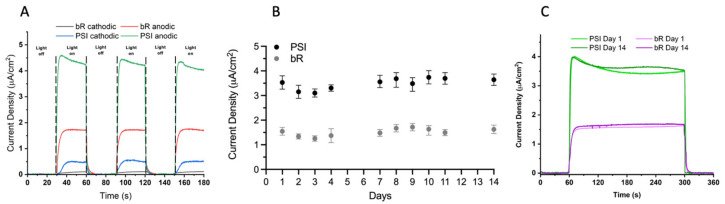
Directional illumination effects on photocurrent density generation and stability of photocurrent response of PSI- and bR-SSCs. (**A**) Comparison of photocurrent densities generated by cathodic or anodic illumination of PSI- or bR-sensitized devices. Traces were baseline corrected using QSOAS software. (**B**) Photocurrent response was measured over two weeks under the same white light illumination regime at 10% light intensity. Each datapoint is the mean photocurrent from three individual traces, and the error bars represent the standard deviation of the mean. (**C**) The average of three individual photocurrent density traces for the PSI-SSC and bR-SSC on days 1 and 14 are shown.

**Table 1 ijms-23-03865-t001:** Properties of the pigment–protein complexes bacteriorhodopsin and photosystem I.

Protein	Pigment	# Pigments per Monomer	MW per Monomer	Abs. Max. (nm)	Ext. Coeff. at Abs. Max. (mM^−1^ cm^−1^)	Organism Source	In Vivo Function
Bacteriorhodopsin (bR)	Retinal	1	27 kDa	560 (green)	63	*Halobacterium salinarum* S9	unidirectional H^+^ pump
Photosystem I (PSI)	Chlorophyll *a*	~100	~400 kDa	680 (red)	57	*Themosynechococcus* *elongatus* BP-1	unidirectional e^−^ transfer

**Table 2 ijms-23-03865-t002:** Electrochemical parameters of aqueous Co^II/III^ redox mediator with varying electrodes.

Electrode Material	*E*_ox_ (V vs. NHE)	*E*_red_ (V vs. NHE)	*E*_m_ (V vs. NHE)
Graphene	0.448	−0.493	−0.045
PEDOT/CNT	0.468	0.003	0.236
Pt NPs	0.221	0.023	0.122
Pt wire	0.173	0.081	0.127
Glassy Carbon	0.205	0.069	0.137

**Table 3 ijms-23-03865-t003:** PSI-sensitized solar cells (PSI-SSC) performance using different counter electrodes and a liquid-based aqueous Co^II/III^ electrolyte.

	*V_OC_* (mV)	*J_SC_* (µA/cm^2^)	FF %	PCE %
PEDOT/CNTs	−132	10.00	25	0.33
Pt NPs	−104	2.22	28	0.06
Graphene	−93	1.30	24	0.03

**Table 4 ijms-23-03865-t004:** PSI-sensitized solar cells (PSI-SSC) performance using liquid and gel-based electrolytes using aqueous-soluble Co^II/III^ redox mediators.

	*V_OC_* (mV)	*J_SC_* (µA/cm^2^)	FF %	PCE %
Gel	−190	9.67	26	0.48
Liquid	−132	9.94	25	0.33

**Table 5 ijms-23-03865-t005:** Bio-sensitized solar cells (BSSC) performance using different proteins.

	*V_OC_* (mV)	*J_SC_* (µA/cm^2^)	FF %	PCE %
bR	−298	13.67	26	1.04
PSI	−259	11.25	24	0.70
Blank	−192	7.44	25	0.35

## Data Availability

The data presented in this study are available on request from the corresponding author. The data are not publicly available due to privacy reasons.

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
