# Peer review of "PEDOT-Carbon Nanotube Counter Electrodes and Bipyridine Cobalt (II/III) Mediators as Universally Compatible Components in Bio-Sensitized Solar Cells Using Photosystem I and Bacteriorhodopsin"

_ijms, 2022, doi:10.3390/ijms23073865_

Round 1

Reviewer 1 Report

The manuscript "PEDOT-Carbon Nanotube Counter Electrodes and Bipyridine Cobalt(II/III) Mediators As Universally Compatible Components in Biosensitized Solar Cells using Photosystem I and Bacteriorhodopsin” is well-structured and very informative. The work continues authors’ research related to the use of PS1 in photoelectrochemical devices, especially utilization of a new redox mediator based on Cobalt bipyridine. The present article is devoted to the use of PEDOT/Carbon nanotubes composite as a counter electrode, as well as electrolyte optimization and also testing the approach using different biological systems.  The work is relevant in a broad global context and written in a simple and clear manner. The design of the experiment is beyond doubt, all the main conclusions are supported by the data obtained. The manuscript will fit well within the scope of the magazine, therefore I will recommend its acceptance. I have only a few comments intended to improve the paper before the publication of the final version.

1) There is inconsistency in tables 3-5. For instance, data in lines “PEDOT/CNT” (Table 3) and “Liquid” (Table 4) in my opinion should match because they describe the same device. However, there are slightly different Jsc values. One can observe even more noticeable differences between the lines “Gel” (Table 4) and “PSI” (table 5) although they again should coincide. Please provide an explanation for these discrepancies.

2) Figure 4 shows that distance between Bipyridine Cobalt(II/III) redox peaks increase with use of PEDOT/CNT electrodes in comparison with Pt, GC and Plastisol. Such behavior usually indicates deterioration of redox reaction kinetics while you claim that use of PEDOT/CNT resulted in an improved cell performance due to reduced charge-transfer resistance. Please clarify this contradiction. It is also necessary to provide some additional experimental details concerning the cyclic voltammetry method, namely working electrode area, Bipyridine Cobalt(II/III) concentration and also supporting electrolyte composition. 

3) Since Bipyridine Cobalt(II/III) is one of the key component of the cell I recommend showing its chemical structure as a separate scheme/figure. Equation 1 does not give a clear understanding of the processes. 

4) I recommend to rewrite heading “2.3 A gel-based electrolyte with aqueous CoII/III redox mediator and PEDOT/CNT counter electrodes may be universally biocompatible in BSSCs” in a more concise manner. 

5) There are a number of typos in the text of the article. For example,  “cobalt (ii/iii)-based…” (Line 315) or a incorrect mention of figure 5 instead of figure 6 (LIne 291). I recommend rechecking the text of the article for such errors.  

Author Response

Response to Reviewer #1

The manuscript "PEDOT-Carbon Nanotube Counter Electrodes and Bipyridine Cobalt(II/III) Mediators As Universally Compatible Components in Biosensitized Solar Cells using Photosystem I and Bacteriorhodopsin” is well-structured and very informative. The work continues authors’ research related to the use of PS1 in photoelectrochemical devices, especially utilization of a new redox mediator based on Cobalt bipyridine. The present article is devoted to the use of PEDOT/Carbon nanotubes composite as a counter electrode, as well as electrolyte optimization and also testing the approach using different biological systems.  The work is relevant in a broad global context and written in a simple and clear manner. The design of the experiment is beyond doubt, all the main conclusions are supported by the data obtained. The manuscript will fit well within the scope of the magazine, therefore I will recommend its acceptance. I have only a few comments intended to improve the paper before the publication of the final version.

  • There is inconsistency in tables 3-5. For instance, data in lines “PEDOT/CNT” (Table 3) and “Liquid” (Table 4) in my opinion should match because they describe the same device. However, there are slightly different Jsc values. One can observe even more noticeable differences between the lines “Gel” (Table 4) and “PSI” (table 5) although they again should coincide. Please provide an explanation for these discrepancies.

Response:  Yes, these represent different experiment with different devices.  The difference in Table 3 vs. 4 is very minor in the Jsc values, this is well within the experimental variability. We actually state this on Page 9 line 341-342:

“The JSC is unchanged as light absorption and transduction into current should not affected by the electrolyte composition”.

  • Figure 4 shows that distance between Bipyridine Cobalt(II/III) redox peaks increase with use of PEDOT/CNT electrodes in comparison with Pt, GC and Plastisol. Such behavior usually indicates deterioration of redox reaction kinetics while you claim that use of PEDOT/CNT resulted in an improved cell performance due to reduced charge-transfer resistance. Please clarify this contradiction. It is also necessary to provide some additional experimental details concerning the cyclic voltammetry method, namely working electrode area, Bipyridine Cobalt(II/III) concentration and also supporting electrolyte composition.

Response:  We are not clear on why the peak distances are more separated in the PEDOT/CNT electrodes nor the graphene, possibly this suggest that electron transfer kinetics are not fast enough at this scan rate to be truly reversible. The morphology of this surface is more complex yet since it consistently performs better in our device we conclude the increase surface area provided by the CNTs also function to reduce the charge-transfer resistance most probably to the Cobalt mediator.  We have included the concentration and the electrolyte composition on Page 15, line 666-667. The area was not explicitly stated but can be seen in Figure 4A.

  • Since Bipyridine Cobalt(II/III) is one of the key component of the cell I recommend showing its chemical structure as a separate scheme/figure. Equation 1 does not give a clear understanding of the processes.

Response: Although this has been shown in our prior publications this is a good suggestion and we have added this as Fig. 5.

  • I recommend to rewrite heading “2.3 A gel-based electrolyte with aqueous CoII/III redox mediator and PEDOT/CNT counter electrodes may be universally biocompatible in BSSCs”in a more concise manner.

Response: We have followed this advice and changed this heading to:

Aqueous CoII/III redox mediator gel electrolytes and PEDOT/CNT counter electrodes are compatible with multiple proteins

5) There are a number of typos in the text of the article. For example,  “cobalt (ii/iii)-based…” (Line 315) or a incorrect mention of figure 5 instead of figure 6 (LIne 291). I recommend rechecking the text of the article for such errors.  

Response: We have checked this again and tried to correct.

Reviewer 2 Report

The authors developed a modified PEDOT electrode system that serves as a counter electrode in a biosensitized PV to improve performance.  A combination of PEDOT with CNTs to prepare a composite was used in this work as counter electrodes. The manuscript can be accepted for publication once the following minor revisions are addressed -

  1. The Edge view image of sintered TiO2 in Fig. 3B needs the layers to be labeled for differentiating the photoanode
  2. The authors are recommended to describe the protein purification process in a small detail under the section ‘Materials and Methods’ instead of directly citing the previously published works
  3. Minor typos in English exist throughout and a review of the manuscript by a native English speaker is recommended

Author Response

The authors developed a modified PEDOT electrode system that serves as a counter electrode in a biosensitized PV to improve performance.  A combination of PEDOT with CNTs to prepare a composite was used in this work as counter electrodes. The manuscript can be accepted for publication once the following minor revisions are addressed -

  1. The Edge view image of sintered TiO2 in Fig. 3B needs the layers to be labeled for differentiating the photoanode

Response:  Yes, this has now been done in White text.

  1. The authors are recommended to describe the protein purification process in a small detail under the section ‘Materials and Methods’ instead of directly citing the previously published works

Response:  We have now added a brief but descriptive outline of the purification procedure on Page 15, line 631-652.

  1. Minor typos in English exist throughout and a review of the manuscript by a native English speaker is recommended

Response:  Thank you we have now proofread by not one but two native English speakers.